# Ultra-Wideband Flexible Absorber in Microwave Frequency Band

**DOI:** 10.3390/ma13214883

**Published:** 2020-10-30

**Authors:** Shicheng Fan, Yaoliang Song

**Affiliations:** Department of Electronic Engineering, Nanjing University Of Science and Technology, Nanjing 210094, China; ylsong@njust.edu.cn

**Keywords:** absorber, broadband, flexible

## Abstract

In this paper, an ultra-wideband flexible absorber is proposed. Based on a summary of the absorption mechanism, using lossless air to replace the heavy lossy dielectric layer will not substantially impact the absorption. The dielectric layer is only a thin layer of polyimide. The proposed absorber is a sandwich structure. The surface is a layer of copper metal ring and wire, and it is loaded with chip resistors to expand the absorber bandwidth. Simulated results show that the bandwidth of the proposed absorber, with an absorptivity of more than 90%, is 2.55–10.07 GHz, with a relative bandwidth over 119.2%. When the electromagnetic wave has a wide incident angle, the absorber still maintains a high absorption. This absorber has been fabricated by FPC (flexible printed circuit) technology. The proposed absorber was attached to the cylinder and measured. The measurement results are roughly the same as the simulation results. The fabricated absorber is easy to carry and flexible, such that it can easily be conformed to irregular objects. The proposed absorber is polarization-insensitive, low profile, thin, and portable, so it is easier to apply in a variety of practical fields.

## 1. Introduction

Landy first proposed perfect absorbers in 2008 [1]. They are different from the Salisbury screen [2] and the Jaumann absorber [3], which require a very thick dielectric layer. Unit cells and dielectric materials can be designed so that incidentally nearly 100% of electromagnetic waves can be absorbed. The metamaterial absorber has a compact structure, a light weight, and an excellent absorption performance, so it has been applied in various fields, such as satellite stealth, RCS (radar cross section) reduction [4,5,6], EMC (electromagnetic compatibility), and EMI (electromagnetic interference) [7]. Broadband capabilities and flexibility make an absorber more practical. Therefore, how to design an ultra-wideband, flexible, metamaterial absorber has become the focus of further practical applications.

Various methods of expanding the absorption bandwidth have been reported. In [8], by using the fractal structure, the structure was optimized to merge three frequency bands, but it was found to be more difficult to further expand the bandwidth by increasing the resonant structure. A broadband absorber can also be achieved by making arrays of differently scaled structures [9]; however, as the number of structures increases, the overall absorption effects of the absorber will decrease. Stacking different resonators with different sizes to obtain a higher bandwidth is also a way to expand the bandwidth, but the resulting thickness is the main limitation of multilayer stacking [10]. Another method to extend the absorption bandwidth is to load suitable chip resistors onto the surface structure [11]. This is because chip resistors can greatly increase the equivalent resistance to increase the absorption bandwidth. However, the multi-layer or multi-structure makes the absorber necessarily a rigid structure and cannot be attached to the curved surface. There is also research on flexible absorbers. Many narrow-band flexible absorbers have been proposed in [12,13,14]. In [15], conductive fibers were used to realize a broadband flexible absorber. In [16], an absorber consisted of resistive films of indium tin oxide, which effectively expanded the absorption bandwidth. The absorption rate of the absorber exceeded 90%, in the frequency range of 6.06–14.66 GHz. The absorber in [17] was inkjet-printed on flexible paper with silver nanoparticle ink. The main disadvantage was that the absorption bandwidth was narrow and easily damaged. A flexible and optically transparent broadband absorber was fabricated by electro hydrodynamics (EHD)-printed technology. The absorber maintained an absorption rate higher than 90% at 73.5–110 GHz in [18]. The absorber in [19] used PI (polyimide) as a dielectric layer to achieve flexibility. These absorbers all used flexible materials as an intermediate layer to achieve cylindrical conformity. However, they were bulky, fragile, and expensive, and thus were not conducive to practical applications.

In this paper, an ultra-band flexible absorber based on loaded chip resistors is proposed. Based on a discussion of the absorbing mechanism, lossless air was used to replace the heavy lossy dielectric layer, and the weight of the absorber was greatly reduced. Concurrently, feasibility analysis was carried out. Through experimental verification, the absorber that we designed maintains a high absorption between frequencies 2.54 GHz and 10.07 GHz. The fractional bandwidth is 119%, which is higher than the previous absorber by comparison. It can easily cover the surface of the cylinder and reduce its RCS. Furthermore, the proposed absorber is low profile and lightweight, which makes it more applicable.

## 2. Design and Simulations

Impedance matching and equivalent circuit are often used to explain the principle of wave absorption. From Formula (Equation 1), we need to match the equivalent impedance of the absorber $Z_{in}$ with the impedance of the free-space $Z_{0}$ as much as possible.
(1)$$\begin{array}{c} \Gamma =\left(Z_{in}-Z_{0}\right)/\left(Z_{in}+Z_{0}\right) \end{array}$$

Therefore, the design of the absorber seeks to solve the problem of impedance matching. For a “sandwich” (metal structure + dielectric + metal structure) absorber, the real part of the equivalent impedance of the absorber should be as close to $Z_{0}$ as possible, and the imaginary part should as close to zero as possible. The absorber element can be modeled as an RLC series circuit according to [20]. The surface metal structure and the underlying metal determine the equivalent capacitance and equivalent inductance, which determines the resonance frequency. According to Formula (Equation 3), the bandwidth can be expanded by increasing the equivalent resistance *r*. Simultaneously, there are multiple resonance frequencies that can be combined to achieve broadband absorption.

In the previous “sandwich” absorber [21,22], the electromagnetic loss is mainly concentrated in the ohmic loss of the surface metal and the dielectric loss of the dielectric layer. The surface metal structure is copper, so the ohmic loss is small; the dielectric loss of this absorber accounts for the main part, and the bandwidth is narrow [23,24]. When the equivalent resistance of the surface structure is greatly increased, the dielectric loss is almost negligible, so replacing the lossy medium with a lossless medium has little effect on the absorption rate. Based on this, we designed the proposed absorber.
(2)$$\begin{array}{ccc} \omega_{0} & = & \frac{1}{\sqrt{LC}} \end{array}$$
(3)$$\begin{array}{ccc} Q & = & \frac{\omega_{0}}{BW}=\frac{1}{r}\sqrt{\frac{L}{C}} \end{array}$$
(4)$$\begin{array}{ccc} Z_{in} & = & \frac{L}{rC} \end{array}$$

The surface structure consists of two parts with a resonant structure. Each part of the resonant structure is composed of a resonant ring, four chip resistors, and metal wires, as shown in Figure 1a. All of the yellow parts in Figure 1 are metallic copper. The thickness is 0.035 mm, and the electric conductivity is 5.8 × $10^{7}$ S/m. The dielectric layer uses polyimide, the relative permittivity is $\varepsilon_{r}=3.5$ mm, and the dielectric loss tangent is tanδ=0.008. The heavy lossy dielectric layer was replaced with lossless air. Between the dielectric layer and the metal bottom plate is a 9.9-mm-thick air layer. The dielectric constant of air is 1. The optimal structural parameters were found by simulating different structural parameters. The structure has the following geometrical parameters: P=20 mm, $L_{1}=13$ mm, $L_{2}=6$ mm, $L_{3}=8.4$ mm, $L_{4}=10$ mm, g=2 mm, t=0.1 mm, H=9.9 mm, and w=0.5 mm. The chip resistance value of the large resonance ring is 91 Ω, and the others are 51 Ω.

The absorber was simulated in CST (2018) Microwave Studio by a frequency domain solver. Unit cell boundary conditions were used in the x and y directions. The incident direction of the electromagnetic wave is along the negative direction of the *Z* axis. The direction of the electric field is the y direction. The direction of the magnetic field is along the positive *x*-axis. According to the formula $A\left(\omega \right)=1-{\left|S_{11}\left(\omega \right)\right|}^{2}-{\left|S_{21}\left(\omega \right)\right|}^{2}$, the absorption rate can be calculated. $S_{11}\left(\omega \right)$$S_{21}\left(\omega \right)$ is the reflection and transmission. Since the ground is a pure metal layer, $S_{21}\left(\omega \right)$ is zero. When we need to calculate the equivalent dielectric constant and equivalent permeability, we need to make a small opening on the bottom metal surface so that $S_{21}$ is not zero.

As shown in Figure 2, the absorber has absorptions peaks at 2.94 GHz, 4.06 GHz, 6.79 GHz, and 9.6 GHz (with peak absorptivity of 99.1%, 99.4%, 96.8%, and 99.8%, respectively). The frequency of the absorption band ranges from 2.55 GHz to 10.07 GHz, including S, C, and X bands. At the same time, the absorption rate of the two resonant rings is also simulated separately. It can be observed in Figure 2 that the large resonance ring has two resonance peaks, and the small one has a resonance peak between the two resonance peaks. When two resonant structures are merged, not only can the three resonant peaks be connected, but one more resonant peak can also be generated. In order to connect the four resonance peaks, the designed structural parameter $L_{3}$ makes the local resonance of the small resonance ring disappear. Due to the chip resistors, the bandwidth of a single resonance peak is increased, and multiple resonance peaks are combined at the same time to achieve ultra-wideband wave absorption.

The loss distribution can be observed by the value of the surface current. To clarify the main loss when absorbing electromagnetic waves, the surface current distributions at peak resonators frequencies 2.91 GHz, 4.06 GHz, 6.79 GHz, and 9.6 GHz are shown in Figure 3. Red and blue are used to indicate loss distribution, and red is used to indicate high loss. It can be clearly seen that the current loss is mainly concentrated in the large square ring and the chip resistor at 2.91 GHz. Most of the current loss is distributed in the small ring and chip resistors at 4.06 GHz, while at 6.79 GHz, the currents are focused in the ancillary structure of the large ring. The two resonant rings are coupled to form the fourth absorption peak at 9.6 GHz. It can be inferred that there should be a fifth resonance peak in the the small square ring part, but not in the broadband, so it is not considered. In Figure 3, it can be seen that the red color is mainly concentrated in the chip resistor part. There is substantial electromagnetic loss when the current passes through the chip resistor. Particularly, at four frequency points, each resonant structure experiences current loss. This is why the absorber can maintain an ultra-wide band.

In order to analyze the absorption mechanism, we used the algorithm in [25,26,27] to calculate the equivalent permittivity, permeability, and relative impedance through S parameters, and the results are shown in Figure 4. It can be seen in the figure that, between 2.4 GHz and 10 GHz, the real part of the relative impedance is close to 1 (approximately equal to the free-space impedance), and the imaginary part is close to 0. As can be seen in Figure 4a, the real part of the equivalent dielectric constant between 2.4 GHz and 10.1 GHz is negative, indicating that the structure has an electrical resonance at this frequency. The imaginary part of the equivalent dielectric constant between 2.4 GHz and 10.1 GHz is positive and maintains a certain value, which means that there is electric field loss. As can be seen in Figure 4b, between the frequency 2.4 GHz and 10.1 GHz, the imaginary part of the equivalent permeability gradually decreases from the maximum value, indicating that the absorption of electromagnetic waves by this structure is based on the combined effect of the magnetic resonance mechanism and the electrical resonance mechanism. This is because, after the chip resistance increases above the equivalent resistance of the surface structure, any surface current caused by electrical resonance and magnetic resonance will produce large losses, so as long as there is resonance, high absorption can be maintained.

In order to be more practical, the absorber generally needs to be polarization-insensitive and have a wide incident angle. The unit cell overlaps itself after rotating 90 degrees. If the surface structure is rotationally symmetric, then the absorber is polarization-insensitive, as shown in Figure 5. The absorption rate hardly changes when the polarization angle is changed from $0^{\circ }$ to $90^{\circ }$. Moreover, the absorption rate of the absorber was simulated at different incident angles with TE and TM polarizations. The electric field direction of the incident wave always remains in the Y-axis direction during TE polarization. In Figure 6a, under TE polarization, as the incident angle increases, the absorption bandwidth decreases, and when the incident angle increases to $60^{\circ }$, the absorption effect becomes worse. In Figure 6b, under TM polarization, as the incident angle increases, the absorption bandwidth is almost unchanged while maintaining high absorption. Only when the incident angle is increased to $60^{\circ }$ does the absorption effect become worse. When the incident angle becomes larger under TE polarization, the coupling degree of the large and small square rings decreases, which directly affects the fourth wave absorption peak. The induced current generated by the magnetic field is the main part of the loss. When the incident magnetic field remains unchanged, a high absorption can be maintained. When the incident angle is too large, the electromagnetic waves passing through the absorber will decrease, which will cause the absorption rate to decrease.

To get closer to the application environment, we need to simulate the absorption rate when the thickness of the air layer changes [28]. The thickness of the air layer will affect the equivalent capacitance, leading to mismatch. The increase or decrease of the thickness of the air layer will cause the mismatch of the absorber. Mismatch will affect the absorption rate of the proposed absorber. The results are shown in Figure 7. However, the results show that, when the thickness of the air layer does not change much, the absorption rate can be kept higher than 90%. This provides the feasibility for the following practical application.

## 3. Measurements Results

The prototype sample shown in Figure 8 was fabricated by printing the surface copper structure on a 0.1 mm PI film. The chip resistor was then soldered to the PI film. Chip resistors were packaged in 0805 (length: 2 mm; width: 1.2 mm). Finally, the film was attached to a copper surface with a size of 20 cm × 20 cm through plastic screws and PI tape. The prototype sample size was 20 cm × 20 cm and contained 10 × 10 unit structures. A 9.9-mm-thick air layer was maintained between the PI film and the underlying metal with the support of sponge tape. Sponge tape will cause errors in the thickness of the air layer. We found, through simulation and analysis, that a certain thickness change does not affect the absorption rate. Polyimide has outstanding heat resistance and mechanical properties, so it is widely used in various fields. The fabricated absorber was cheap, easy to make, and very flexible. The measurement environment is shown in Figure 9. In order to make the measurement more accurate and ensure the far-field distance, the distance between the antenna and the sample was greater than 2 m. In the microwave anechoic chamber, two double-ridged horn antennas (1–18 GHz) and a vector network analyzer (PNA-XN5244A) were used to measure the S parameters of the proposed absorber. A reference object was fabricated, and a piece of rubber with a size of 20 cm × 20 cm was covered with copper foil. First, the S parameter of the reference object was measured. The S parameter of the absorber was then measured while keeping the proposed absorber in the same location and condition. By subtracting the two sets of measurement data [29], the interference caused by many factors, such as edge reflection, diffraction loss, and scattering loss, was reduced.

The measurement result is shown in Figure 10, and is in good agreement with the simulation. The measured absorption rate is not as smooth as the simulation result. It can be seen that the measured absorption rate fluctuates, which is caused by the lack of complete agreement that resulted when the reference object was used to measure the data. Fabrication tolerances and a limited number of elements are also reasons for the difference between the measurement results and the simulation results. The chip resistors were welded manually, and the position was not accurate.

Since the designed absorber is flexible, it is of course necessary to measure its effect in actual application, so we covered the absorber on a copper cylinder with a diameter of 15 cm to measure the absorber effect. Figure 11 shows the measured absorptivity when the designed absorber is attached to the cylinder. With the measurement results, we can conclude that the designed absorber can be effectively used on many occasions to reduce the RCS of the object, regardless of whether the object is flat or not.

## 4. Conclusions

In summary, an ultra-wideband flexible absorber with polarization-insensitive, wide-angle incidence characteristics was investigated. By using flexible printed circuit (FPC) technology to fabricate and chip resistors to expand bandwidth, the proposed absorber has an ultra-wide band and is flexible, to a degree that far exceeds other absorbers, as shown in Table 1. This can be achieved because the designed absorber has multiple resonant structures on a single-layer medium loaded with resistance simultaneously. The simulated result shows that the absorber has a relative bandwidth of 119%, and the absorber frequency band includes the S, C, and X bands. The proposed absorber can also be used at higher frequencies when the package of the chip resistor is changed from 0805 to 0402 (length: 1 mm; width: 0.5 mm). Due to the geometrical scalability, the absorbing frequency can be changed from 5 GHz to 18 GHz. The designed absorber has been attached to the cylinder and measured. The measurement results are roughly the same as the simulation results. The main body of the proposed absorber consists of a PI film with a thickness of 0.1 mm, so it is light, small in size, flexible, and thin, making it suitable for portable applications. The proposed absorber is expected to be applied in many fields, such as radar target stealth, satellite stealth, and electromagnetic protection.

## Figures and Tables

**Figure 1 materials-13-04883-f001:**
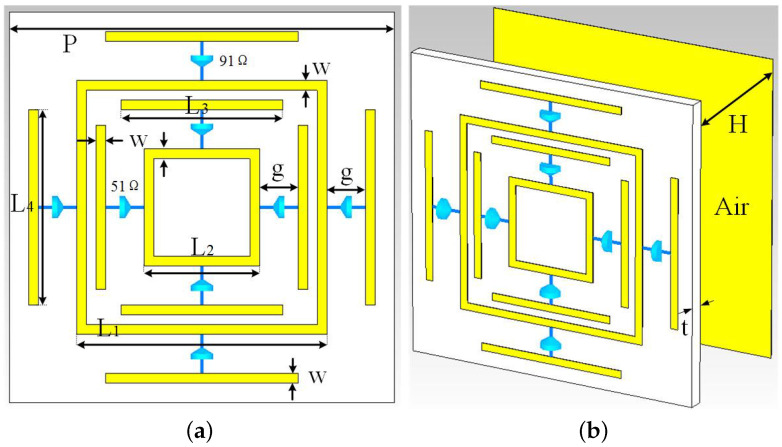
Unit structure parameters of the designed absorber. (**a**) Front view; (**b**) Oblique view.

**Figure 2 materials-13-04883-f002:**
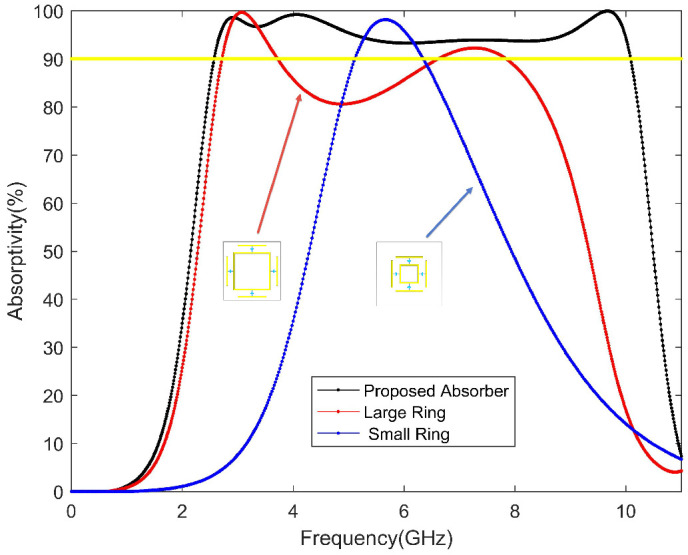
Absorption results of the proposed absorber.

**Figure 3 materials-13-04883-f003:**
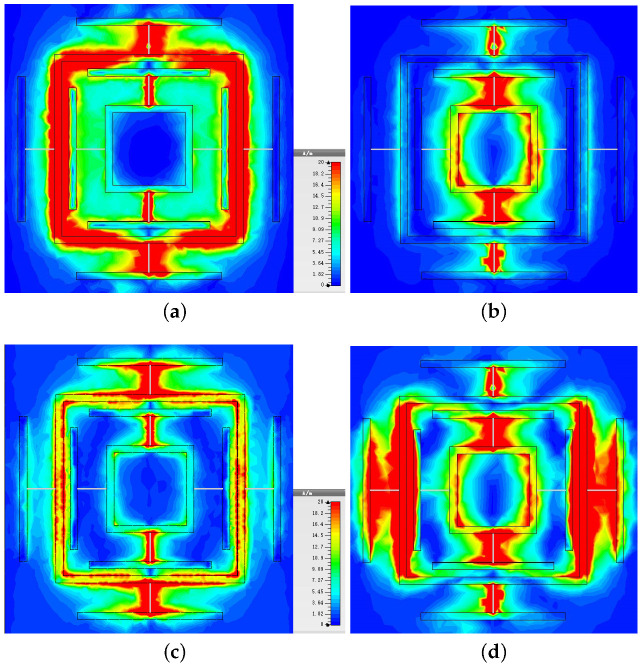
Surface current loss distribution of the proposed absorber, at the absorption peaks of (**a**) 2.91 GHz, (**b**) 4.06 GHz, (**c**) 6.79 GHz, and (**d**) 9.6 GHz.

**Figure 4 materials-13-04883-f004:**
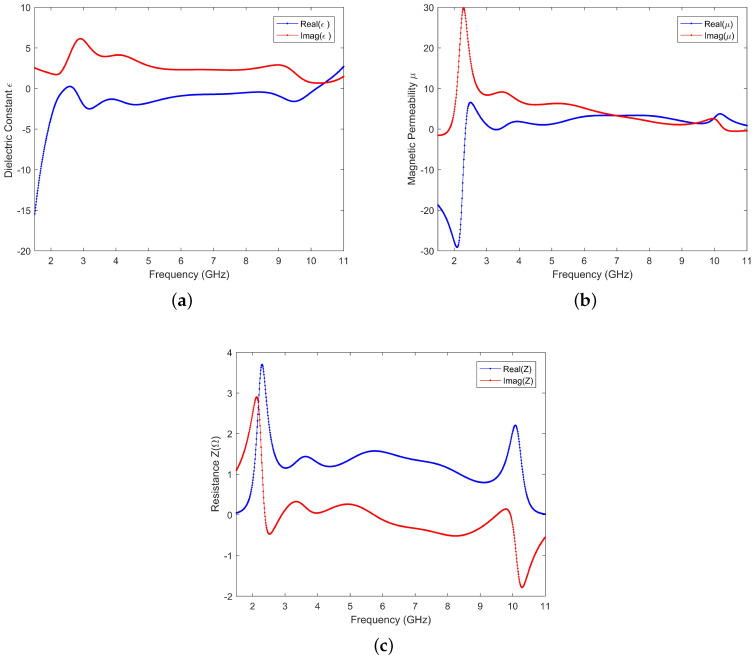
(**a**) Equivalent dielectric constant. (**b**) Equivalent permeability. (**c**) Relative resistance calculated by electromagnetic inversion.

**Figure 5 materials-13-04883-f005:**
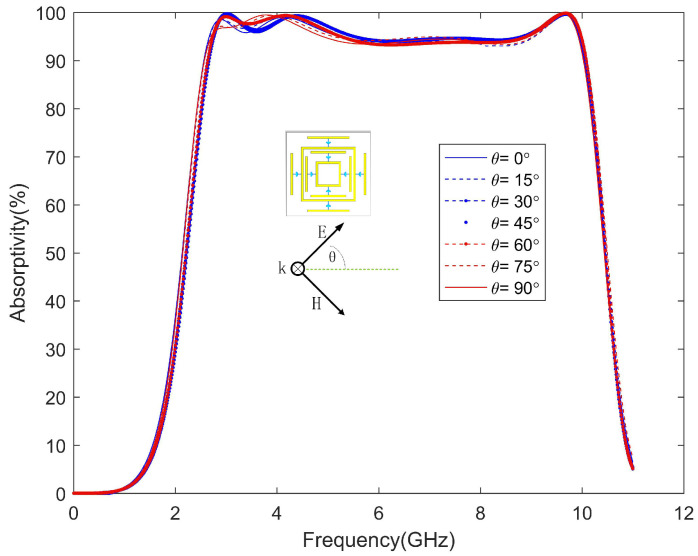
Simulated absorptivity for different polarization angles.

**Figure 6 materials-13-04883-f006:**
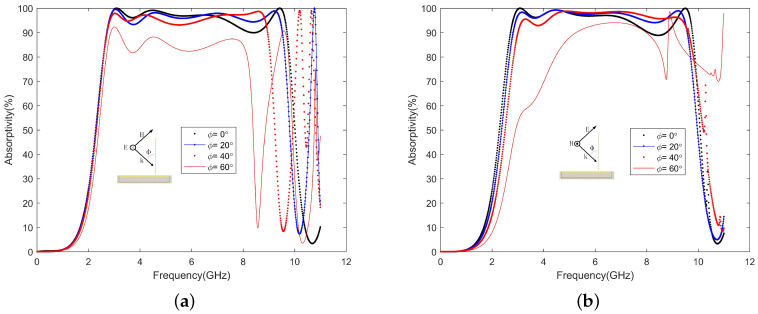
Different incidence angles for (**a**) TE and (**b**) TM polarizations.

**Figure 7 materials-13-04883-f007:**
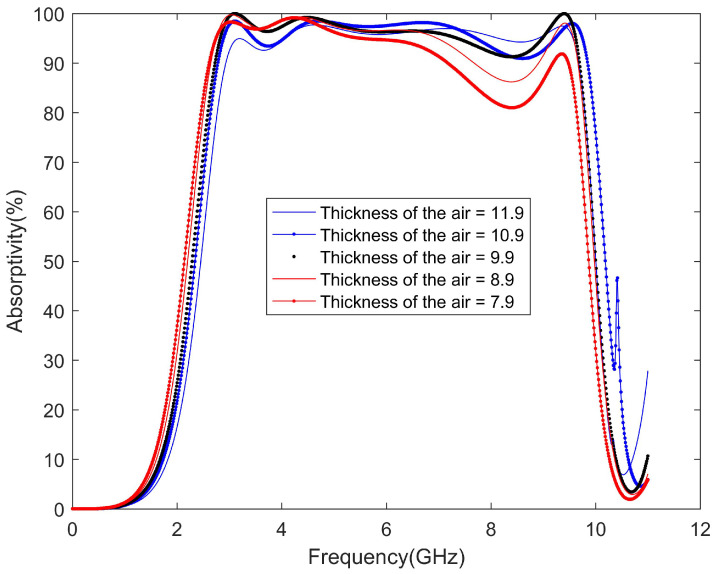
Simulated absorptivity for different thicknesses of the air layer.

**Figure 8 materials-13-04883-f008:**
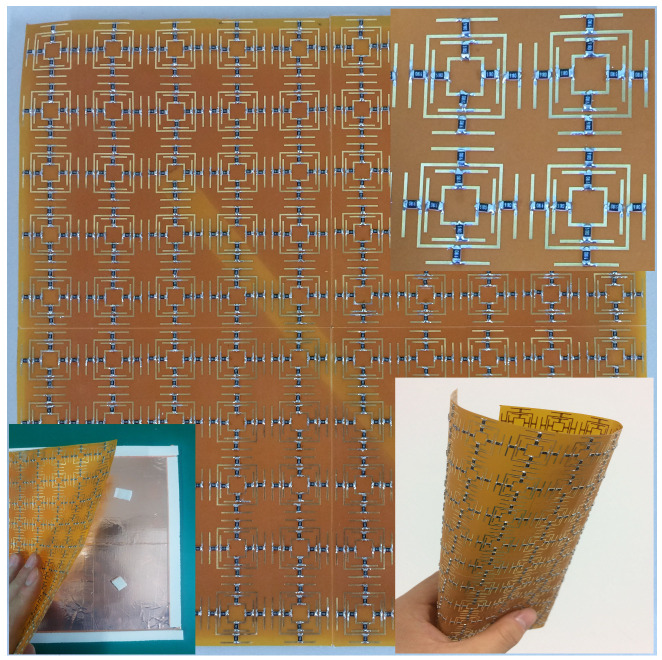
Photograph of the fabricated absorber.

**Figure 9 materials-13-04883-f009:**
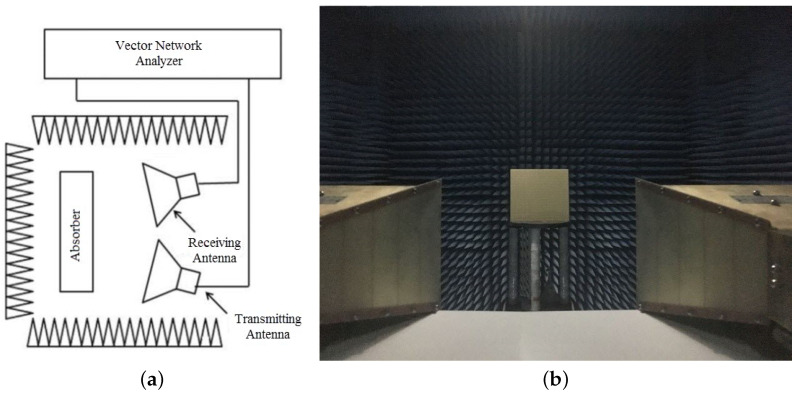
(**a**) Schematic diagram of measurement environment (**b**) Actual measurement environment.

**Figure 10 materials-13-04883-f010:**
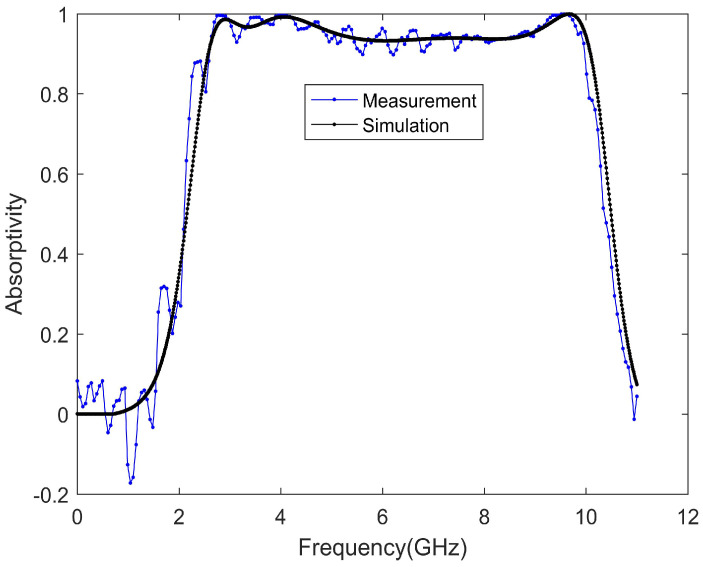
Comparison of the absorption rate between simulation and measurement.

**Figure 11 materials-13-04883-f011:**
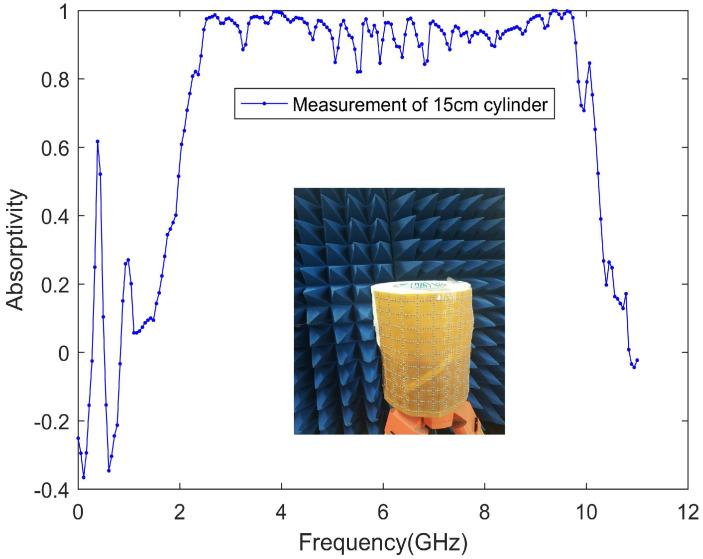
Measurement of the absorption rate attached to the cylinder.

**Table 1 materials-13-04883-t001:** Comparison With Previous Absorbers. Fractional bandwidth $=2*\left(f_{H}-f_{L}\right)/\left(f_{H}+f_{L}\right)*100\%$.

Absorber	Fractional Bandwidth	Flexible
[30]	117%	No
[31]	70.8%	No
[32]	7.2%	Yes
[33]	18.9%	Yes
[34]	82.6%	Yes
Proposed	119%	Yes

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
