# Peer review of "Ultra-Wideband Flexible Absorber in Microwave Frequency Band"

_materials, 2020, doi:10.3390/ma13214883_

Round 1

Reviewer 1 Report

The authors report about a flexible absorber in the Microwave band which represent an interesting topic.

However, it is not clear to me in the abstract and in the introduction why the authors said about removing an unimportant dielectric layer if the metal ring and wire are patterned on polyimide layer. Which is the removed layer?

Overall, the manuscript needs to be improved in terms of reading.

I will suggest accepting with major revision if the following issue will be addressed.

  • Which is the unimportant dielectric layer?
  • Improve the reference to literature adding for instance some ref at line 20, 63, 66, 135 and other.
  • Improve caption and legend in all figures
  • Insert a 3D image of Figure 1 to better understand the design top-bottom
  • In lien 155, which is the reference object?
  • How the authors calculates the bandwidth?

Author Response

Dear reviewer

On behalf of my co-authors, we thank you very much for giving us an opportunity to revise our manuscript. We appreciate editors and reviewers very much for their positive and constructive comments and suggestions on our manuscript entitled “Ultra-wideband Flexible Absorber In Microwave Frequency Band” (ID materials-975888). We have carefully studied reviewer’s comments and tried revised our manuscript according to the comments. The modification is marked in YELLOW in the revised manuscript.

We would like to express our great appreciation to you and reviewers for comments on our paper. Looking forward to hearing from you.

Thank you and best regards.

Yours sincerely

Shicheng Fan

Oct 27th, 2020

Reviewer 2 Report

The manuscript presents a flexible absorber in the microwave regime. By using a multi-resonant structure loaded with chip resistors, ultrawideband absorption was achieved. With absorption bandwidth much larger compared to previously reported absorbers and simple flexible design, current work makes an important contribution to the field of microwave absorbers. However, the manuscript can be significantly improved before it can be published. I recommend minor revisions to the manuscript. Below are some comments/suggestions to further improve the manuscript for better understanding and readability.

Comments:

  • The language of the manuscript requires significant improvement. Various sentences in the manuscript are not clear and require rephrasing, tense correction, spelling corrections etc. I would recommend the manuscript be proofread by a native English speaker before resubmission.

Some examples: Line21 “ultra-band” ; Line 23 “many literatures” and others

  • Line 20, Abbreviations such as RCS, EMI , EMC etc. should be defined in their first use.
  • Line 19-20. More references in support of the statements should be provided.
  • Line 24. Provide reasons for why it is difficult to increase bandwidth in this case?

Line 33. “ …There has not been much research on flexible metamaterial absorbers…” The statement is not  correct. There are many works in the literature on this (see below) and therefore the authors should modify this statement and should list more suitable references on flexible absorbers and mention the challenges faced by them.

Zhang et al. Journal of Alloys and Compounds, 705, 262-268, 2017.

Chejarla et al.  Electronics Letters,  55, 3, 133-134, 2019.

Huijie Chen et al. Mater. Res. Express, 5, 015804, 2018.

  • Line 45. This is the main crux of the manuscript. Mention exactly the underlying principle for absorption, and its consequences on bandwidth and other parameters and how this is better than other works.
  • Line 47. Mention the quantitative value of absorption and how much larger is this compared to previous works.
  • Line 56. “What is “sandwich” absorber. Explain clearly.
  • The chip resistors, metal wire etc. should be clearly indicated in figure 1.
  • Line 71: Mention the units of thickness.
  • Line 73. What does the statement “ Use lossless media instead of lossy media.” Mean?
  • The structure uses air layer. To make this clear in the figure 1b, add a label for dielectric constant of air.
  • The label “p” is not clear. Is it the width of the metal wires and the resonant ring?
  • Line 76. line width is 0.5mm. What do you mean by linewidth? Not clear.
  • It would be nice to label the resistance values in the figure 1a.
  • How were the geometrical parameters chosen. Were they optimized through simulation. This should be described clearly in the text.
  • The figure caption of figure 2 should be more elaborate for clarity of the figure. What are the various colored graphs represent?
  • The scientific reasoning for the appearance of two or single peaks in the resonant rings should be explained. Why does the larger ring have two resonance peak compared to the smaller one. Were the chip resistors included in the simulation of individual resonators?
  • The authors should explain how the current loss was calculated to obtain figure 3. The units and the colorbar should be mentioned
  • While the method to calculate the equivalent permittivity, permeability from the S parameters might be straightforward, it should be mentioned in the text or appropriate reference to calculate these parameters should be mentioned.
  • Line 109-115. References to appropriate figure parts a, b etc. should be mentioned for clarity and easy understanding.
  • Line 113. “ and maintains a certain height”. What does this mean. The authors should explain physical effects of  positive imaginary coefficient.
  • Line 121. “ The surface structure is rotationally symmetric” .Why do the authors say the structure is rotationally symmetric. From the figure the structure is axially symmetric but not rotational. I would expect absorption to change between 0 and 90 degrees.
  • Line 144. Chip resistors are packaged in 0805. What is 0805? Explain for clarity.
  • Line 145. Mention units wherever necessary.
  • The height was maintained using sponge tape. The authors should show a side view of the device showing the air gap as this is related with the major claims of the manuscript. How easy is it to maintain the gap since it is flexible substrate, I would assume the gap to vary at different positions. Add some comments about this and how it can impact the performance or alternate methods to improve the stability.
  • The experimental set up should be described in more detail. A block diagram with all the components should be mentioned for clarity of the readers.
  • How was the air gap with the copper cylinder maintained during the measurements?
  • Table 1. Define fractional bandwidth. How it is calculated?

Author Response

Dear reviewer

On behalf of my co-authors, we thank you very much for giving us an opportunity to revise our manuscript. We appreciate editors and reviewers very much for their positive and constructive comments and suggestions on our manuscript entitled “Ultra-wideband Flexible Absorber In Microwave Frequency Band” (ID materials-975888). We have carefully studied reviewer’s comments and tried revised our manuscript according to the comments. The modification is marked in YELLOW in the revised manuscript.

We would like to express our great appreciation to you and reviewers for comments on our paper. Looking forward to hearing from you.

Thank you and best regards.

Yours sincerely

Shicheng Fan

Oct 27th,2020

Round 2

Reviewer 1 Report

The authors addressed all the requested comments, therefore I would suggest to accept the manuscript for publication.